# Utility of the Diffusion Weighted Sequence in Gynecological Imaging: Review Article

**DOI:** 10.3390/cancers14184468

**Published:** 2022-09-15

**Authors:** Apurva Bonde, Eduardo Andreazza Dal Lago, Bryan Foster, Sanaz Javadi, Sarah Palmquist, Priya Bhosale

**Affiliations:** 1Department of Radiology, The University of Texas Health Science Center at San Antonio, San Antonio, TX 78229, USA; 2Department of Radiology, The University of Texas MD Anderson Cancer Center, Houston, TX 77030, USA; 3Department of Radiology, Oregon Health & Science University, Portland, OR 97239, USA

**Keywords:** gynecological cancers, imaging, diffusion weighted imaging

## Abstract

**Simple Summary:**

Diffusion weighted imaging (DWI) is a magnetic resonance imaging sequence with diverse clinical applications in malignant and nonmalignant gynecological conditions. It provides vital supplemental information in the diagnosis and management of various gynecological conditions. Radiologists should be aware of fundamental concepts, clinical applications and pitfalls of DWI. Additionally we briefly discuss potential scope of newer advanced techniques based on DWI including diffusion tensor imaging and diffusion-weighted whole-body imaging with background signal suppression.

**Abstract:**

Functional imaging with diffusion-weighted imaging (DWI) is a complementary tool to conventional diagnostic magnetic resonance imaging sequences. It is being increasingly investigated to predict tumor response and assess tumor recurrence. We elucidate the specific technical modifications of DWI preferred for gynecological imaging, including the different b-values and planes for image acquisition. Additionally, we discuss the problems and potential pitfalls encountered during DWI interpretation and ways to overcome them. DWI has a wide range of clinical applications in malignant and non-malignant gynecological conditions. It provides supplemental information helpful in diagnosing and managing tubo-ovarian abscess, uterine fibroids, endometriosis, adnexal torsion, and dermoid. Similarly, DWI has diverse applications in gynecological oncology in diagnosis, staging, detection of recurrent disease, and tumor response assessment. Quantitative evaluation with apparent diffusion coefficient (ADC) measurement is being increasingly evaluated for correlation with various tumor parameters in managing gynecological malignancies aiding in preoperative treatment planning. Newer advanced DWI techniques of diffusion tensor imaging (DTI) and whole body DWI with background suppression (DWIBS) and their potential uses in pelvic nerve mapping, preoperative planning, and fertility-preserving surgeries are briefly discussed.

## 1. Introduction

Diffusion-weighted imaging (DWI) is an essential MRI technique in evaluating the female pelvis, routinely used to identify and potentially characterize both benign and malignant gynecologic diseases [1]. Initially, the DWI technique was used in brain imaging, especially to detect a small and early acute ischemic stroke. Currently, DWI is widely used in abdominal imaging [2,3].

In the last few decades, the advent of parallel imaging and rapid data acquisition has significantly improved the quality of the DWI technique. This has made its use possible in several clinical applications, providing high-quality exams with a better signal-to-noise ratio (SNR) without significantly increasing the total scan time duration [2,4].

DWI is also widely used in gynecological conditions to detect disease and assess response to treatment. It helps detect and characterize benign lesions, such as tubo-ovarian abscesses and endometriomas, and malignant lesions, such as endometrial, cervical, and ovarian cancers. Other critical applications include identifying peritoneal disease, assessing tumor response following therapy, evaluating tumor recurrence, and differentiating malignant lesions from post-radiation fibrosis [1,4,5]. This article reviews the basics of DWI techniques, pitfalls, and major clinical applications, and discusses the future trends in diffusion-weighted imaging in gynecological entities.

## 2. Principles of DWI and Image Acquisition Technique

DWI is a functional MRI technique that enables evaluation of tissues and their microenvironment by dephasing and rephasing water molecules, dispensing with the need for contrast media. It is based on the free, constant, and random movement of intra- and extracellular water molecules (Brownian movement). In tissue, the motion of water molecules can be decreased (restricted diffusion) or increased (free diffusion) depending on tissue type, intra- and extracellular interactions, and distinct biological situations (e.g., inflammation, ischemia) [6].

DWI is a T2-weighted fast spin-echo sequence after applying two equal and opposite sensitizing gradients before and after the 180^o^ radiofrequency pulse. If the tissue has restricted diffusion, the second gradient cancels the first, resulting in a hyperintensity appearance on DWI. On the other hand, if tissue has low cellularity and/or defective cell membranes, a relatively higher extracellular space allows water molecules to move more freely and randomly, leading to a lower signal on DWI. Those gradients are acquired with different strengths, characterized by the sensitizing fields amplitude, duration, and temporal spacing, which reflect different b-values [2].

Typically, pelvic MRIs are performed with at least one low b-value (0–50 s/mm^2^) and one or more intermediate to high b-values (≥600 s/mm^2^). At low b-value acquisitions, highly mobile intravascular water molecules within vessels will have a signal loss, so-called “black-blood”. In contrast, at higher b-values, tissues characterized by hypercellularity (e.g., malignant tumors) show restricted diffusion [7]. Besides malignant neoplasms, some normal hypercellular organs that can show restricted diffusion are endometrium during secretory-phase, ovaries, lymph nodes, and bowel mucosa. Other materials showing restricted diffusion include blood, melanin, and iron. The mobility of water molecules can also be impeded in high-protein content, viscous cystic lesions (e.g., abscesses), and tissues with ample connective tissue matrix, such as fibrosis, resulting in restricted diffusion [2].

Apparent diffusion coefficient (ADC) is a quantitative measure of tissue diffusivity expressed in ×10^−3^ as mm^2^/s and derived from the slope of exponential decrease in signal between at least two DWI b-values. The accuracy of ADC maps depends on the number of b values acquired, misregistration between different b-value acquisitions, and signal-to-noise ratio (SNR) [4]. Ideally, multiple b-values would be acquired. However, this significantly increases imaging time, and in clinical practice, it is most common to acquire no more than three b-values [4].

Major limitations of DWI are related to the large gradients that make this technique sensitive to microscopic diffusion, which also makes them susceptible to macroscopic motion and susceptibility artifacts, which are magnified at 3T compared to 1.5T MR imaging. Those artifacts, frequently encountered on pelvic bowel loops, can be minimized by shorter echo times (TE), smaller numbers of echo train lengths (ETLs), and wider receiver bandwidth [8]. There are also limitations related to low spatial resolution and low SNR [7].

The DWI acquisition for the female pelvis includes multiple b-values (e.g., 0, 500, 1000) and planes. The axial plane is commonly used for imaging. Sometimes additional or modified axial planes can be acquired as they can provide additional helpful information depending on the disease, such as sagittal and axial oblique planes in endometrial cancer and axial oblique plane in cervical cancer. The evaluation should always be correlated with conventional anatomic sequences [3].

## 3. Clinical Applications in Gynecological Non-Malignant Conditions

### 3.1. Uterine Fibroids

Fibroids have variable composition depending upon the amount of fibrosis and smooth muscle and may undergo various types of degeneration and hence show varying degrees of diffusion restriction (Figure 1 and Figure 2). European society of urogenital radiology (ESUR) 2018 guidelines suggest DWI to be used as an optional MRI sequence for imaging of fibroids providing supplemental information [9]. Cellular fibroids with compact smooth muscle fibers show diffusion restriction analogous to their typical T2-hyperintense appearance. Typical T2-dark fibroids and some degenerated fibroids show inhomogeneous or absent diffusion restriction (Figure 3). Although diffusion restriction can be seen in cellular fibroids and uterine leiomyosarcoma (Figure 4), ADC values are lower in uterine leiomyosarcoma than in cellular fibroids [1]. Different cut off for ADC values, for example, less than 1.1 × 10^−3^ mm^2^/s or less than 1.08 × 10^−3^ mm^2^/s, were investigated and proposed for aiding the differentiation between cellular uterine fibroid and leiomyosarcoma and were found to be somewhat useful [10,11]. However, there has been no definite cut off for ADC value due to the significant overlap between fibroids and leiomyosarcoma. Given the overlap of MRI imaging features of leiomyosarcoma and fibroids, a combination of conventional and DWI imaging features are to be particularly useful to avoid overlooking the malignant sarcomas for benign fibroids, and DWI improves accuracy of MRI to 92% [12,13]. MRI algorithm has been proposed using high DWI signal as compared to endometrium, low ADC values, and lymphadenopathy, and was found to be useful for differentiating leiomyosarcoma and uterine fibroids [14].

Additionally, DWI was used for monitoring of the response of fibroids to treatment, particularly when no significant change in size was noted after treatment (Figure 5). An increase in ADC values after treatment of uterine fibroids can be seen due to tissue necrosis [1].

### 3.2. Adnexal Torsion

Adnexal torsion can lead to hemorrhagic infarction of the ovaries and fallopian tubes due to compromised blood supply [15]. Ovarian torsion can be challenging to diagnose sonographically, particularly when the clinical picture is indeterminate. In such cases, MRI is useful [15]. A torsed ovary demonstrates enlargement and lack of enhancement with ipsilateral uterine deviation on conventional MRI [1]. DWI may help support the detection and diagnosis of ovarian/fallopian tube torsion. It can show diffusion restriction in the torsed ovary, fallopian tube, or both due to hemorrhagic infarction and associated cytotoxic edema [1,15] (Figure 6). DWI can be particularly helpful in detecting an abnormal ovary when used in non-contrast large field-of-view MRI protocols for abdominal pain in children and pregnant women. Both normal and abnormal ovaries stand out from background suppressed tissues on high b-value DWI, which allows the reader to not only localize the ovaries easily but evaluate symmetry. However, on DWI, there is significant overlap with other benign and malignant entities, and diagnosis requires not only a high degree of suspicion but recognition of typical findings on other sequences (enlarged ovary, peripheral follicles, stromal edema, whirlpool sign, etc.).

### 3.3. Endometriosis

Endometriomas can show a variable degree of diffusion restriction related to their internal blood and hemosiderin content and should not be confused with malignant entities (Figure 7). Therefore, rare cases of malignant transformation of endometriomas can be challenging to diagnose based solely on diffusion restriction. Evaluation of conventional imaging characteristics such as thickened septae, and T2 hyperintense and enhancing mural nodules, as well as metastatic disease aid in diagnosing rare cases of malignant transformation of endometrioma and its differentiation from benign endometrioma showing diffusion restriction. Additionally, DWI can assist in the differentiation from hemorrhagic cysts as endometriomas tend to have lower and inhomogeneous ADC values compared with hemorrhagic cysts [16,17,18]. Balaban et al. reported mean ADC values of (1.15 ± 0.2) × 10^−3^ mm^2^/s for endometriomas and (2.10 ± 0.1) × 10^−3^ mm^2^/s for hemorrhagic ovarian cysts at a b value of 1000 [17]. However, other signs, such as the T2 dark spot sign and lack of T2 shading provide a more definitive differentiation between the hemorrhagic cyst and endometrioma [15].

### 3.4. Tubo-Ovarian Abscess

The imaging appearance of the tubo-ovarian abscess (TOA) depends upon its internal content. Diffusion restriction is commonly seen due to its highly viscous internal proteinaceous content compromised of bacteria, debris, inflammatory cells, and necrotic tissues (Figure 8 and Figure 9). However, the internal content has been observed to change depending on the chronicity of the TOA; small, chronic abscesses or TOA after antibiotic treatment may not show diffusion restriction (Figure 10). Hence, variable or absence of diffusion restriction may be seen depending upon internal content and acuity/chronicity of the abscess. DWI is particularly helpful in diagnosis when IV contrast cannot be used for computed tomography (CT) or MRI [19]. As both malignancies and TOAs can show diffusion restriction, the clinical presentation is essential in differentiation [20].

### 3.5. Mature Cystic Teratoma

Ovarian mature cystic teratoma is the most common benign ovarian tumor in younger females, consisting of tissues of at least two of three embryonic layers (mesoderm, ectoderm, and endoderm). Diffusion restriction can be present due to internal keratinous material and fat or sebum-containing fluid [1,15] (Figure 11 and Figure 12). Malignant transformation of dermoid rarely occurs in 1–2% of cases and can also show diffusion restriction. Intermediate to high T2 signal intensity and avid enhancement of an invasive mass extending beyond the cyst wall helps differentiate malignant transformation from benign ovarian dermoid cyst as both can show diffusion restriction [15].

### 3.6. Ovarian Fibroma, Fibrothecoma, and Thecoma

In most cases, ovarian thecoma, fibroma, and fibrothecoma show intermediate signal intensity on DWI compared to myometrium [21,22]. Seventy-three percent of patients with thecoma, fibroma, and fibrothecoma showed low signal intensity on DWI due to the T2 blackout effect described subsequently in this article, and a higher number of fibroma in a study by Chung et al. suggested that the combined use of DWI and conventional magnetic resonance imaging (MRI) sequences helps differentiate ovarian fibroma, fibrothecoma, and thecoma from ovarian malignancies [21]. However, because of their high stromal proliferation, few ovarian fibromas, fibrothecomas, and thecomas may show diffusion restriction [15,19]. Therefore, in general, DWI is not helpful in characterizing these tumors.

## 4. Clinical Applications in Gynecological Malignancies

The addition of DWI to conventional MRI sequences aids in the diagnosis and detection of female pelvic malignancies, and locally recurrent and metastatic disease, and provides useful information for tumor response assessment and differentiation from post-radiation fibrosis [23,24]. Standardization of DWI techniques with ADC values may be helpful in the evaluation of response but need validation in future prospective trials [25,26]. DWI has shown 90–95% accuracy for detecting peritoneal metastatic deposits in gynecological malignancy [1,5].

### 4.1. Cervical Cancer

DWI plays a complementary role in diagnosing and staging cervical cancer as cancer tissues and metastatic lymph nodes show restricted diffusion with lower ADC values, aiding in their characterization with increased confidence [25,27] (Figure 9 and Figure 13). Cervical cancer is typically well delineated as a T2 hyperintense lesion in the T2 hypointense cervical stroma. However, DWI is helpful in younger patients when the cervical stroma is T2 hyperintense or when the tumor is very small and difficult to visualize on T2-weighted images [7]. DWI also helps to avoid overestimating the extent of cervical cancer due to inflammation or edema seen on T2-weighted images [1]. Additionally, DWI has improved accuracy in detecting parametrial invasion and metastatic lymph nodes, as aggressive tumors with parametrial invasion and lymph node involvement have lower ADC values than those without parametrial invasion and lymph node involvement [27]. When used with other conventional sequences, DWI improves performance and increases confidence in tumor detection and accurate measurement [27]. Additionally, lower ADC values are noted in squamous cell carcinoma histology, high-grade tumors, as well as recurrent and metastatic disease and, therefore, may predict poor survival [27].

DWI helps predict the tumor response to chemoradiation in cervical cancer as ADC values are potential biomarkers for tumor response assessment [28] (Figure 14). Tumors with internal necrosis or cystic areas with higher ADC values indicate more hypoxic tumor tissue and are less responsive to chemoradiation. Tumors with low pretreatment ADC values are more responsive to chemoradiation [29]. Additionally, DWI is also helpful for determining tumor response to chemoradiation in cervical cancer when morphological changes are not apparent on other sequences, as tumors show an increase in ADC value from baseline. Persistent lower ADC values in the tumor after chemoradiation are associated with incomplete or poor treatment response [7]. However, a meta-analysis by Meyer et al. showed that pretreatment ADC values alone do not help predict response to chemoradiation [30]. Mean ADC values have been shown to predict the risk of recurrence before cervical cancer therapy [27,31].

DWI helps differentiate posttreatment changes from the residual tumor as the residual tumor has lower ADC values than posttreatment changes, particularly in the case of cervical cancer [32].

Advanced imaging techniques based on DWI, including texture analysis with ADC map, and ADC histogram analysis, are under investigation for their clinical use in research studies. Texture analysis with ADC map random forest model is found to be better than ADC values in research studies for assessment of important prognostic factors of cervical cancer, improving diagnostic performance for noninvasive evaluation of tumor grade, histological type, parametrial invasion, lymph nodal metastatic involvement, staging, and recurrence and prediction of recurrence-free survival and risk stratification, which is helpful for proper selection of high-risk patients who would benefit from more aggressive treatment, at the same time avoiding overtreatment of low-risk patients [33]. Pretreatment ADC histogram analysis using 3D lesion region of interest (ROI) for ADC measurements in cervical cancer is better in identifying patients with a high risk of recurrence and thus can predict disease-free survival [34]. These patients may be candidates for aggressive therapy. Diffusion maps obtained from the intravoxel incoherent model (IVIM) is a promising tool for determining tumor aggressiveness and predicting tumor response to the treatment of cervical cancer [35]. Post-radiation increase of MMP-9 expression in cervical cancer is associated with increased ADC values, indicating that ADC can be used as a biomarker for predicting tumor response assessment of cervical cancer [36]. Texture analysis with DWI has the potential to predict recurrence in low-volume cervical cancers [37]. However, further studies are required to establish standardized guidelines for its use in regular clinical work.

### 4.2. Endometrial Cancer

Given the high DWI signal of endometrial cancer and hypointensity of adjacent myometrium, DWI has also been useful for accurate determination of the depth of myometrial invasion [1]. The combined use of T2-weighted and DWI images increases the sensitivity for accurately detecting myometrial invasion from endometrial cancer [38]. DWI improves the sensitivity of detection of metastatic lymph nodes by using the size and ADC values in conjunction, and it is comparable to dynamic contrast-enhanced MRI for detecting nodal metastasis from endometrial cancer in some studies and [39,40]. Axial planes for DWI acquisition are commonly used; however, the oblique axial plane can be acquired, especially when cervical invasion by endometrial cancer is suspected, and an additional sagittal plane improves the radiological assessment of endometrial cancer, particularly for the determination of the extent of myometrial invasion [41,42,43].

DWI has a role in differentiating benign from malignant endometrial lesions. Benign endometrial lesions, including endometrial polyp, endometrial hyperplasia, and endometrial thickening due to physiological etiologies, show statistically significant less diffusion restriction with higher ADC values than endometrial cancer (Figure 15). Mean ADC values with endometrial cancer were (0.8 ± 0.1) × 10^−3^ mm^2^/s, while that of benign endometrial conditions were (1.4 ± 0.2) × 10^−3^ mm^2^/s [44]. For proper measurement of ADC values, the ROI should be placed to avoid any areas of necrosis and cystic components of the lesion to avoid false elevation of values [44]. Three dimensional radiomics using DWI can help identification of endometrial cancer patients with high-risk histopathological factors such as advanced stage and deep myometrial invasion [45]. However, the study by Bereby-Kahane et al., reported limited value of radiomics analysis based on T2-weighted and DWI MRI [46]. Quantitative biomarkers using DWI, dynamic contrast MRI, and IVIM refine diagnostic performance of MRI for risk stratification of endometrial cancer [47]. Another study by Liu et al., demonstrated that radiomics based on multiparametric MRI including DWI is helpful in predicting progression-free survival in endometrial cancer patients [48].

### 4.3. Ovarian Cancer

DWI has a relatively limited role in differentiating benign and malignant ovarian lesions (Figure 16, Figure 17 and Figure 18). Several benign ovarian lesions or conditions described above may show diffusion restriction similar to malignant lesions [15,19]. When the solid component of the ovarian lesion is hypointense on T2-weighted and high b value DWI images, the lesion is likely to be benign and categorized in ORADS category 2. There is an increased likelihood of malignancy when there is a T2 intermediate signal with an increased signal on high b-value DWI corresponding to higher ORADS categories 3–5 [49,50]. Multiparametric MRI with DWI and DCE MRI have higher accuracy for differentiating between benign and malignant ovarian lesions [51]. DWI, ADC maps, and conventional MRI sequences improve sensitivity, specificity, and accuracy for detecting ovarian cancer and its nodal and metastatic peritoneal disease [24]. In serous ovarian cancer, significant correlation was found between DWI parameters and tumor biomarkers Ki67 and VEGF predicting the treatment-free survival. In this study by Dolata et al., poorly differentiated epithelial ovarian cancers have lower ADC values and higher Ki67. Additionally, significant negative correlation was noted between ADC values and high-grade versus low-grade epithelial ovarian cancers [52]. Diffusion coefficient (Dk) and volume transfer constant (K^trans^) generated from histogram analysis using advanced DWI and dynamic contrast-enhanced sequence are found to be useful for differentiating borderline and malignant epithelial ovarian tumors [53].

## 5. Pitfalls of DWI

DWI has several pitfalls and limitations. Since DWI is intrinsically a T2-weighted sequence, the DWI signal is affected by the T2 signal. T2 shine-through effect and T2 blackout effects are the two most commonly encountered in clinical practice [1].

### 5.1. T2 Shine-Through Effect

T2 shine-through effect is described when lesions appear hyperintense on both DWI images and ADC maps (Figure 19). This effect is mainly seen in cystic lesions with higher T2 relaxation times and low b-values. It can be avoided by always correlating DWI images with the corresponding ADC map and using higher b-values (>800) for imaging of the female pelvis [7].

### 5.2. T2 Blackout Effect

When a lesion appears hypointense on both DWI and ADC maps, it is described as the T2 blackout effect. Usually, this is seen in benign conditions and occurs due to internal calcifications within the lesion and fibrotic tissues.

### 5.3. Diagnostic Pitfalls

Well-differentiated tumors may show a lack of restricted diffusion or relatively less restricted diffusion due to low cellularity or mixed cystic structure. A benign condition such as hemorrhage and normal structures including ovaries, secretory endometrium, lymph node, and bowel mucosa may show diffusion restriction [1,7]. This can be overcome by looking at the anatomical images. Cystic lesions with internal hemorrhagic, proteinaceous, mucinous, and keratinous content may result in diffusion restriction.

When reproducibility of ADC values and standardized uptake value (SUV) generated from positron emission tomography (PET) scan was compared in epithelial ovarian cancers, good reproducibility was noted on baseline studies; however, lower reproducibility was noted with ADC values as compared with SUV values after treatment with chemotherapy [54].

## 6. DWIBS and DTI

Diffusion-weighted whole-body imaging with background signal suppression (DWIBS) is a new DWI technique that creates 3D PET-like images of the whole body without radiation exposure or the need for contrast media [55]. The DWIBS technique acquires multiple thin slice DWI images using the free-breathing technique and use of STIR-EPI fat suppression; the resultant suppression of background signal from normal organ tissues allows the detection of distant metastasis in oncologic patients [55]. However, many benign lesions and typical structures such as brain, salivary glands, tonsils, spleen, gallbladder, small intestine/small intestinal contents, colon, adrenal glands, prostate, and gonads may show restricted diffusion and, therefore, anatomic sequences are also needed for correlation. DWIBS has been shown to be reasonably accurate for preoperative assessment of ovarian cancer and for detection of unresectable sites of metastatic disease such as mesenteric and large bowel carcinomatosis [56]. While DWIBS may be an alternative imaging modality to PET-CT for detecting lesions throughout the body, there is currently limited data to recommend its use outside of clinical trials [7].

Diffusion tensor imaging is a newer DWI-based method used to estimate the directionality and strength of water diffusion in different tissue structures [57,58]. ADC values generated from diffusion tensor imaging (DTI) (ADC^T^ value) and functional anisotropy (FA) values are commonly used DTI parameters for evaluating the uterus. ADC^T^ values are correlated with the directionality of water molecules in a three-dimensional state and are lower in malignant lesions and higher in benign lesions. FA values indicate directionality of water molecules ranging from 0 to 1. When FA values are close to zero, it is isotropic dispersion, and when they are close to one, it is anisotropic dispersion. Alteration in tissues with disease leads to altered water diffusion. Hence, DTI can calculate diffusion of water molecules in the tissues, aiding in differentiation between normal and diseased tissues [58].

DTI has been used and is a well-established technique for the diagnosis of a wide variety of neurological diseases [59] and can also be used in imaging peripheral nerves such as the sacral roots [60]. It is based not only in the free diffusion rate of water molecules but also in the direction and speed of the water movement that can be measured with the fractional anisotropy (FA) index and illustrated with the 3D reconstruction method of the nerve tracts called tractography. DTI can evaluate microarchitectural abnormalities in sacral roots in patients with endometriosis-associated pain [61], and it correlates with the type of pain and adhesions [62], emerging as a promising tool for personalized therapeutic planning.

DTI has shown a promising role in differentiating degenerated uterine fibroid and uterine sarcoma, which is difficult with conventional imaging due to their overlapping imaging findings. Anisotropy with higher FA values is seen with degenerated fibroids, while lower FA values are noted with uterine sarcomas [58]. Various DTI parameters are also being investigated for usefulness in the radiological assessment of gynecological lesions. ADC values generated from DTI are found to differentiate between normal, benign, and malignant endometrial conditions, increasing the diagnostic confidence of conventional MRI and DWI, while FA values can have a role in predicting the extent of myometrial invasion and tumor grading. DTI has the potential to provide noninvasive information about tumor invasion, tumor grade, and metastatic lymph node detection of cervical cancer [63]. DTI with L5-S1 roots FA cut off value of >0.3099 DTI have demonstrated an accuracy of 73% in the diagnosis of parametrial invasion by cervical cancer in the study by Paola [64]. Cervical DTI can be used to measure microstructural differences in collagen fiber organization and hydration in early and late pregnancy and could be helpful in predicting cervical ripening noninvasively [57].

## 7. Conclusions

DWI is a complementary MRI sequence that provides useful information. It has a wide range of applications in malignant and non-malignant gynecological conditions, playing an essential role in diagnosing or assisting in the diagnosis and providing supplementary information useful for their management. It is instrumental in problem-solving when intravenous contrast is contraindicated. The radiologist must understand the basic concepts of DWI as well as pitfalls and mimics. Novel techniques with diffusion-weighted whole-body imaging with background signal suppression and diffusion tensor imaging show promising potential for the future.

## Figures and Tables

**Figure 1 cancers-14-04468-f001:**
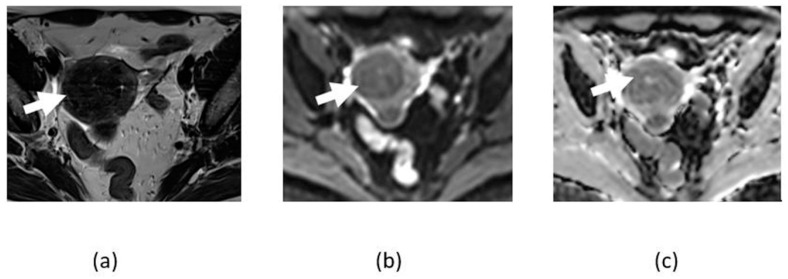
Axial T2-weighted image (**a**) showing hypointense lesion in the uterus fundus consistent with leiomyoma (arrows). The lesion shows low signal on DWI (**b**) and ADC map (**c**), consistent with absence of restricted diffusion.

**Figure 2 cancers-14-04468-f002:**
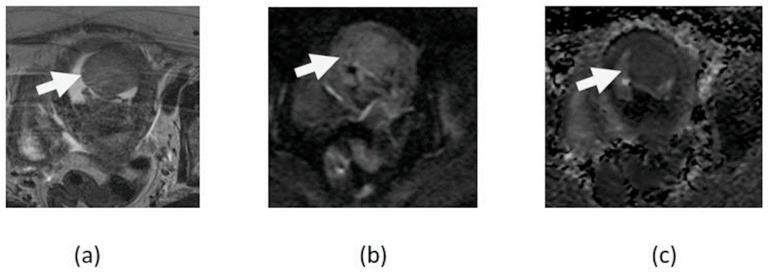
Axial T2-weighted (**a**), DWI (**b**), and ADC map (**c**) image of the pelvis shows T2 intermediate signal submucosal fundal lesion showing restricted diffusion characterized by a high signal on DWI and low signal on ADC map, pathologically proven cellular leiomyoma (arrows).

**Figure 3 cancers-14-04468-f003:**
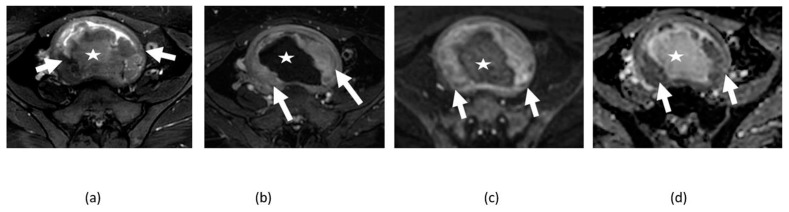
Axial T2-weighted (**a**), contrast-enhanced T1-weighted (**b**), DWI (**c**), and ADC map (**d**) image shows a large T2 heterogenous uterine fibroid with peripheral cellular area showing diffusion restriction (arrows) and central T2 hyperintense non-enhancing area without diffusion restriction in the degenerated portion (asterisk) of the fibroid.

**Figure 4 cancers-14-04468-f004:**
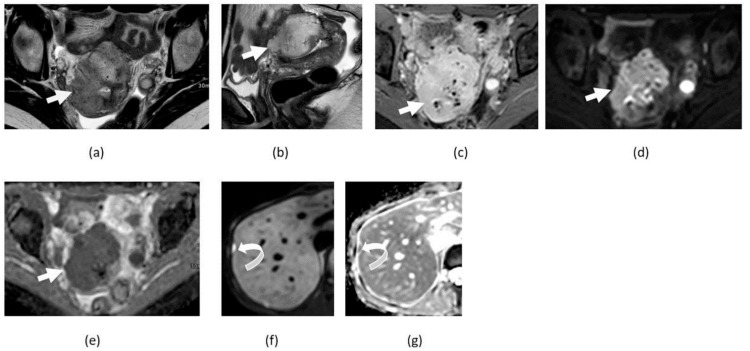
Axial (**a**) and sagittal (**b**) T2-weighted, axial contrast-enhanced T1-weighted (**c**), DWI (**d**), and ADC map (**e**) images of the pelvis showing bulky irregular enhancing anterior uterine wall mass (arrows) with high signal on DWI and marked low signal ADC map consistent with pronounced restricted diffusion from biopsy proven as uterine leiomyosarcoma. Axial DWI (**f**) and ADC map (**g**) of upper abdomen image of the same patient shows a same pattern of restricted diffusion on the capsular liver metastatic lesion (curved arrow).

**Figure 5 cancers-14-04468-f005:**
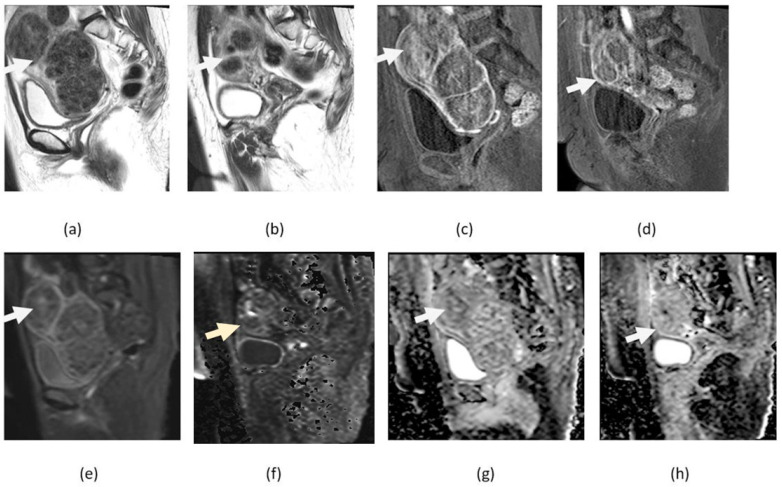
Multiple myometrial nodules consistent with leiomyomas on sagittal T2-weighted images (**a**,**b**) and sagittal contrast-enhanced fat-suppressed T1-weighted (**c**,**d**), before and after treatment with embolization, some fibroids identified with arrows. After treatment, there is a decrease in size and absence of diffusion restriction shown in the sagittal DWI (**e**,**f**) and ADP maps (**g**,**h**), findings that are consistent with treatment response.

**Figure 6 cancers-14-04468-f006:**
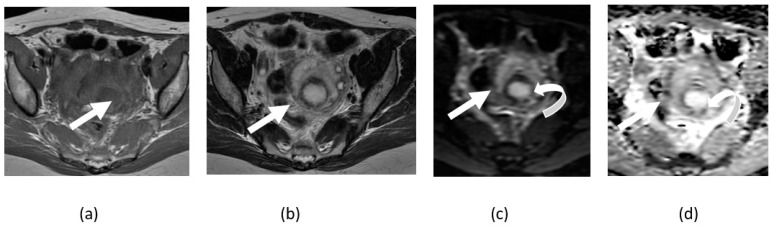
Axial T1- (**a**) and T2-weighted (**b**) images showing heterogeneous left adnexal twisted structure (arrows) with surrounding rim, which shows faint rim of subtle diffusion restriction (curved arrows) on corresponding axial DWI (**c**) and ADP maps (**d**).

**Figure 7 cancers-14-04468-f007:**
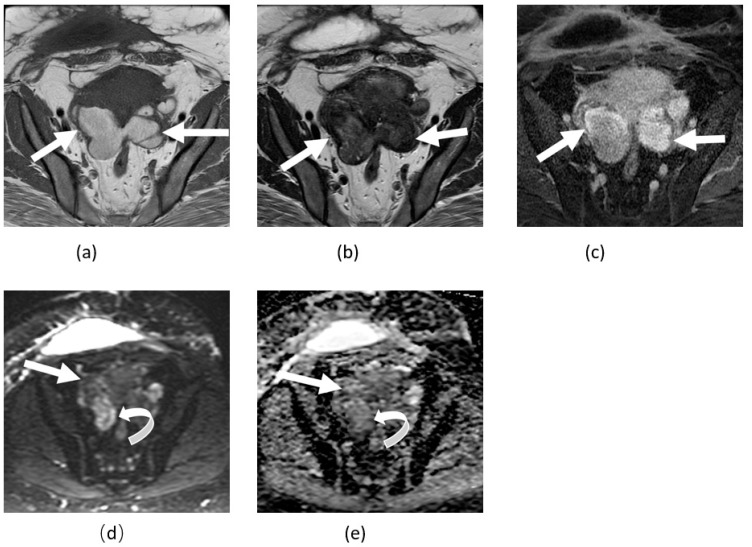
Axial T1 (**a**), T2 (**b**), and T1-weighted unenhanced fat saturation (**c**) images show bilateral enlarged ovarian hemorrhagic cystic lesions consistent with endometriomas (arrows) touching each other in posterior midline of the pouch of Douglas. Axial DWI (**d**) and ADP maps (**e**) nicely depict the variable degree of diffusion restriction of these endometriomas related to different ages of the blood content. Please note right lateral portion of these lesions shows mild diffusion restriction (curved arrows).

**Figure 8 cancers-14-04468-f008:**
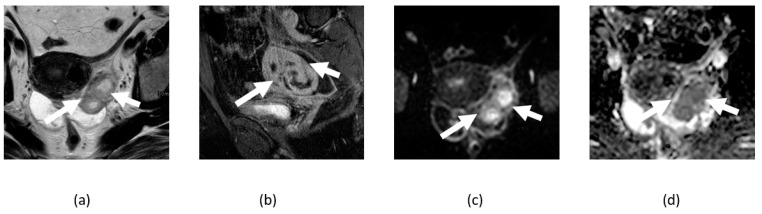
Axial T2-weighted (**a**), sagittal T1-weighted fat saturation post-contrast (**b**), DWI (**c**) and ADP maps (**d**) show serpiginous thick enhancing walled left adnexal pyosalpinx with restricted diffusion (arrows).

**Figure 9 cancers-14-04468-f009:**
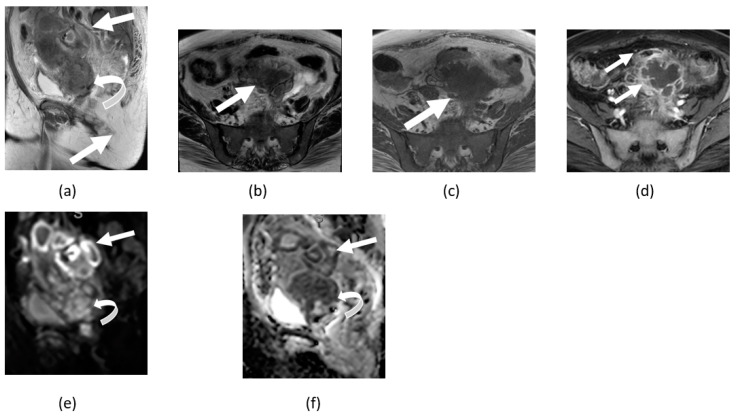
Sagittal (**a**) and axial (**b**) T2-weighted images, axial T1-weighted images (**c**), contrast-enhanced fat-suppressed T1-weighted images (**d**) show mixed signal intensity upper left pelvic rim-enhancing complex collection/tubo-ovarian abscess (straight arrows). Concurrent T2 hypointense cervical mass proven to be cervical cancer (curved arrows) is noted; sagittal T2-weighted (**a**), DWI (**e**) and ADC map (**f**) images show nearly homogenous diffusion restriction. However, please note the peripheral rim-like heterogeneous diffusion restriction in tubo-ovarian abscess in the superior left pelvis (straight arrows).

**Figure 10 cancers-14-04468-f010:**
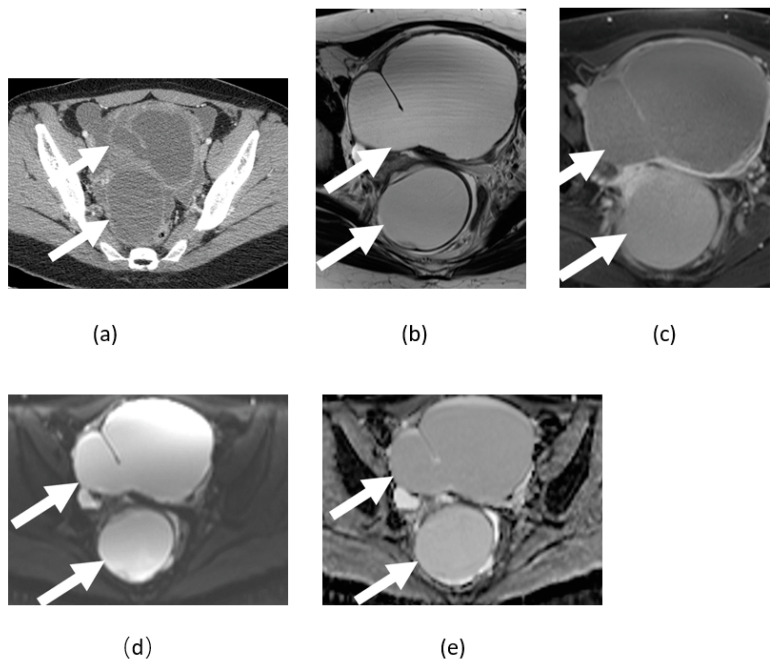
Axial contrast-enhanced CT (**a**), T2- (**b**), and T1-weighted post-contrast (**c**) MRI image of pelvis shows large chronic tubo-ovarian abscess (arrows) without diffusion restriction on axial DWI (**d**) and ADC maps (**e**).

**Figure 11 cancers-14-04468-f011:**
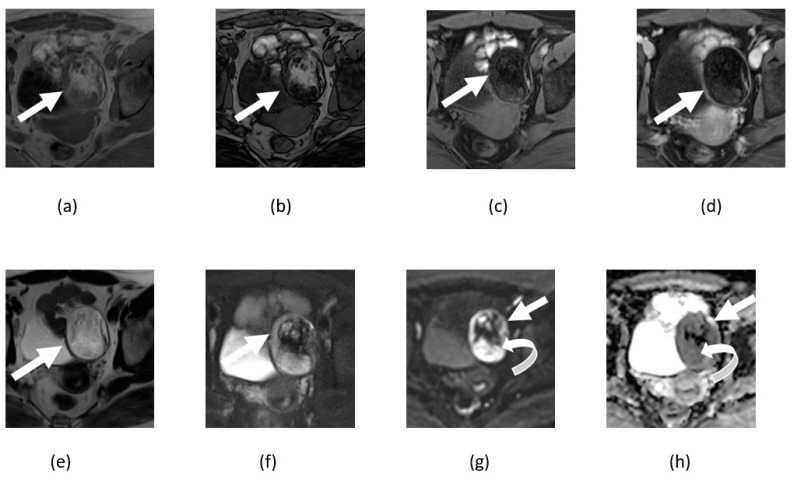
Axial T1-weighted in (**a**) and out (**b**) phase, T1-weighted pre- (**c**) and post-contrast (**d**), and T2-weighted non fat suppressed (**e**) and T2 weighted fat suppressed (**f**) images shows fat containing left adnexal lesion compatible with a mature cystic teratoma (straight arrow). Corresponding DWI (**g**) and ADP maps (**h**) show peripheral component with restricted diffusion (curved arrow).

**Figure 12 cancers-14-04468-f012:**
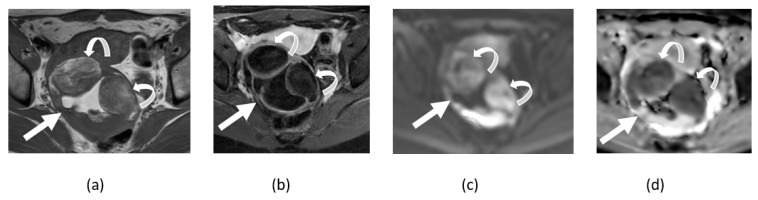
Axial T1-weighted image of pelvis without (**a**) and with (**b**) fat suppression shows fat-containing pelvic lesion consistent with mature cystic teratoma (arrows) showing diffusion restriction in anterior fat-containing component (curved arrows) on corresponding DWI (**c**) and ADC map (**d**) images.

**Figure 13 cancers-14-04468-f013:**
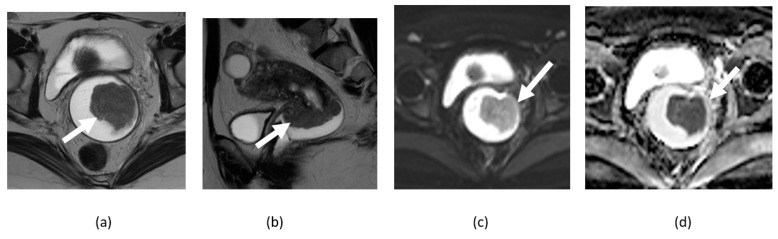
Axial (**a**) and sagittal (**b**) T2-weighted images with vaginal gel shows a large exophytic cervical mass (arrows) from cervical cancer with diffusion restriction on corresponding diffusion weighted image (**c**) and ADC map (**d**).

**Figure 14 cancers-14-04468-f014:**
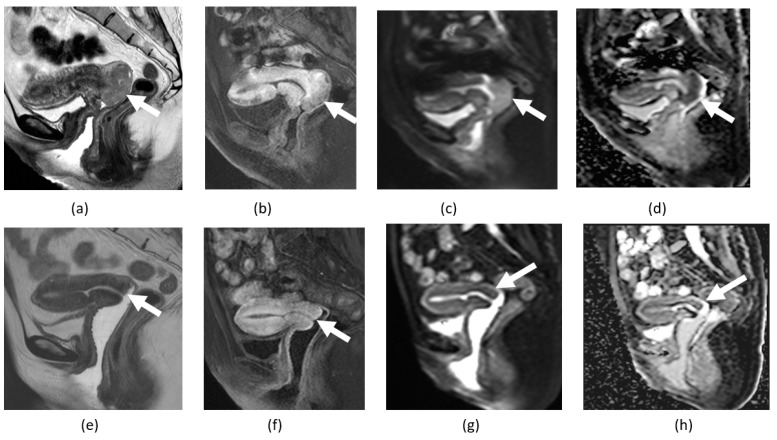
Sagittal T2- (**a**) and T1-weighted contrast-enhanced (**b**) images before treatment shows a large cervical mass (arrows) protruding into superior aspect of vaginal canal and exhibiting restricted diffusion on corresponding DWI (**c**) and ADC maps (**d**). Sagittal T2- (**e**) and T1-weighted contrast-enhanced (**f**) images after treatment shows resolution of the lesion (arrows) and absence of areas with restricted diffusion suggesting no residual viable tumor on corresponding DWI (**g**) and ADC maps (**h**).

**Figure 15 cancers-14-04468-f015:**
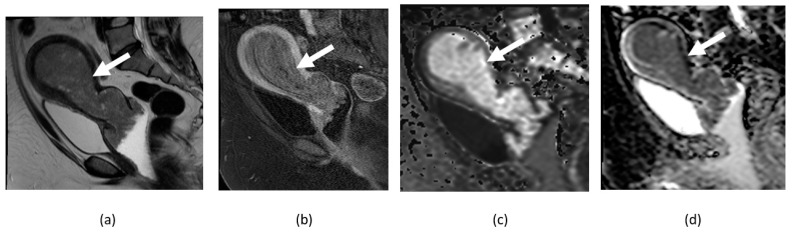
Sagittal T2-weighted (**a**), contrast enhanced T1-weighted (**b**), DWI (**c**), and ADC map (**d**) show a large intracavitary endometrial mass (arrows) with cervical invasion from endometrial cancer showing restricted diffusion.

**Figure 16 cancers-14-04468-f016:**
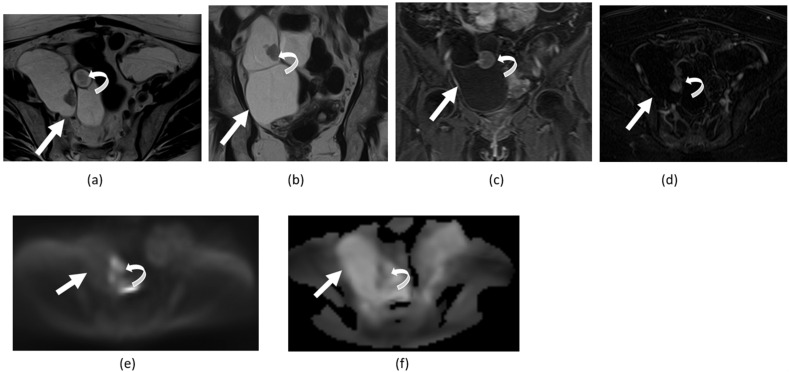
Axial (**a**) and coronal (**b**) T2-weighted images and contrast-enhanced fat-suppressed T1-weighted coronal image (**c**) and axial contrast-enhanced fat-suppressed T1-weighted with subtraction (**d**) show a large multiloculated ovarian mass (straight arrows) with solid component characterized by papillary projections and mural nodules (curved arrows) demonstrating diffusion restriction on DWI (**e**) and ADC maps (**f**). The lesion was pathologically proven to be a cystic ovarian cystadenocarcinoma.

**Figure 17 cancers-14-04468-f017:**
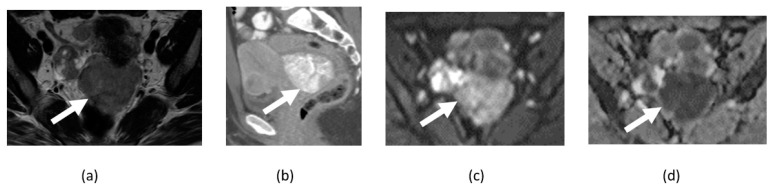
Axial T2-weighted image (**a**) and sagittal contrast-enhanced CT image (**b**) show posterior pelvic lobulated solid mass (arrows) from biopsy serous ovarian tumor with diffusion restriction on corresponding DWI (**c**) and ADC maps (**d**).

**Figure 18 cancers-14-04468-f018:**
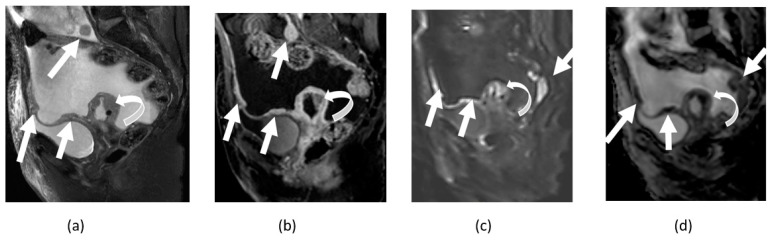
Sagittal T2-weighted (**a**) and T1-weighted FS contrast-enhanced (**b**) images of patient with metastatic endometrial cancer with malignant ascites showing peritoneal thickening and nodules showing diffusion restriction in a small endometrial mass from biopsy-proven endometrial cancer (curved arrows) and metastatic peritoneal nodules (arrows) on the sagittal DWI (**c**) and ADC maps (**d**).

**Figure 19 cancers-14-04468-f019:**
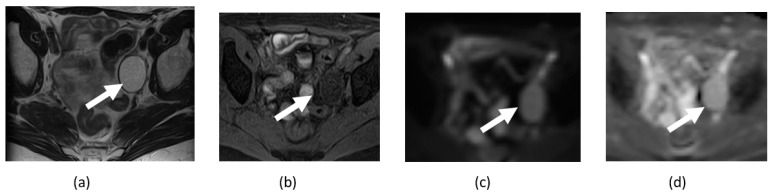
Axial T2-weighted image of the pelvis (**a**) showing hyperintense homogenous ovarian lesion (arrows) with low signal on T1-weighted image with fat suppression (**b**). The lesion shows high signal on low b value DWI (**c**) without the corresponding low signal on ADC map (**d**), suggesting a phenomenon known as “T2 shine-through”.

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
