# Peer review of "Utility of the Diffusion Weighted Sequence in Gynecological Imaging: Review Article"

_cancers, 2022, doi:10.3390/cancers14184468_

Round 1
Reviewer 1 Report
In the review, the authors discuss the use of diffusion-weighted magnetic resonance imaging (DW-MRI) in the elective field of gynecological imaging. In general, this a well written review article that encompass the current applications of DW-MRI.
I have the following comments and suggest a few amendments.
The authors should focus their effort on three points: (i), an update of the references; (ii), introduce the concept of IVIM and DTI, which are both part of DW-MRI and have already some applications and interesting results in the field of gynecological imaging; and (iii), a better and more careful presentation of the figures.
In general, the authors should use major and recent references and add a substantial number of references to update their review.
As minor comments (but to consider)
1. Simple summary: Missing. Please add.
2. Key words: Missing, please add Keywords using MeSH index terms.
Abstract.
3. Dermoid is too vague.
4. Spell out ADC.
Main text.
Introduction
5. Line 41-42. Relevant references are missing after this sentence “It helps detect and characterize benign lesions, such as tubo-ovarian abscesses and endometriomas, and malignant lesions, such as endometrial, cervical, and ovarian cancers.
6. Lines 56-63. The second section of “Principles of DWI” has no references at all. The authors should have relevant references.
In the section “Principles of DWI”
7. The authors should briefly describe IVIM and diffusion tensor imaging which are part of DW-MRI and add the following recent references
Dolciami M, Capuani S, Celli V, Maiuro A, Pernazza A, Palaia I, Di Donato V, Santangelo G, Rizzo SMR, Ricci P, Della Rocca C, Catalano C, Manganaro L. Intravoxel Incoherent Motion (IVIM) MR Quantification in Locally Advanced Cervical Cancer (LACC): Preliminary Study on Assessment of Tumor Aggressiveness and Response to Neoadjuvant Chemotherapy. J Pers Med 2022;12(4):638. doi: 10.3390/jpm12040638.
Di Paola V, Perillo F, Gui B, Russo L, Pierconti F, Fiorentino V, Autorino R, Ferrandina G, Valentini V, Scambia G, Manfredi R. Detection of parametrial invasion in women with uterine cervical cancer using diffusion tensor imaging at 1.5T MRI. Diagn Interv Imaging 2022;doi: 10.1016/j.diii.2022.05.005.
8. In the section titled Clinical applications, the authors should add the following two references for further discussion
Abdel Wahab C, Jannot AS, Bonaffini PA, Bourillon C, Cornou C, Lefrère-Belda MA, Bats AS, Thomassin-Naggara I, Bellucci A, Reinhold C, Fournier LS. Diagnostic Algorithm to Differentiate Benign Atypical Leiomyomas from Malignant Uterine Sarcomas with Diffusion-weighted MRI. Radiology 2020;297(3):E347.
Lin Y, Wu RC, Huang YL, Chen K, Tseng SC, Wang CJ, Chao A, Lai CH, Lin G. Uterine fibroid-like tumors: spectrum of MR imaging findings and their differential diagnosis. Abdom Radiol 2022;47(6):2197-2208. doi: 10.1007/s00261-022-03431-6.
In the section “Cervical carcinoma”
9 The authors should discuss the role of radiomics from ADC maps and add the following reference
Yamada I, Oshima N, Miyasaka N, Wakana K, Wakabayashi A, Sakamoto J, Saida Y, Tateishi U, Kobayashi D. Texture Analysis of Apparent Diffusion Coefficient Maps in Cervical Carcinoma: Correlation with Histopathologic Findings and Prognosis. Radiol Imaging Cancer 2020 ;2(3):e190085. doi: 10.1148/rycan.2020190085.
In the section “Endometrial cancer”
10. In this section, the authors should discuss the application of DW-MRI for determining lymphovascular space invasion (LVSI)
The authors should add the following recent reference
Lefebvre TL, Ueno Y, Dohan A, Chatterjee A, Vallières M, Winter-Reinhold E, Saif S, Levesque IR, Zeng XZ, Forghani R, Seuntjens J, Soyer P, Savadjiev P, Reinhold C. Development and Validation of Multiparametric MRI-based Radiomics Models for Preoperative Risk Stratification of Endometrial Cancer. Radiology 2022; doi: 10.1148/radiol.212873.
11. They should also discuss that one study did not find added value for DW-MRI in the prediction of high grade and LVSI compared to T2-weighted images and add the following reference
Bereby-Kahane M, Dautry R, Matzner-Lober E, Cornelis F, Sebbag-Sfez D, Place V, Mezzadri M, Soyer P, Dohan A. Prediction of tumor grade and lymphovascular space invasion in endometrial adenocarcinoma with MR imaging-based radiomic analysis. Diagn Interv Imaging 2020;101(6):401-411. doi: 10.1016/j.diii.2020.01.003.
12. They should also discuss with more details the role of DW-MRI for determining tumor aggressiveness, myometrial invasion and progression free survival with the following three references
Satta S, Dolciami M, Celli V, Di Stadio F, Perniola G, Palaia I, Pernazza A, Della Rocca C, Rizzo S, Catalano C, Capuani S, Manganaro L. Quantitative diffusion and perfusion MRI in the evaluation of endometrial cancer: validation with histopathological parameters. Br J Radiol 2021;94(1125):20210054. doi: 10.1259/bjr.20210054.
Deng L, Wang QP, Chen X, Duan XY, Wang W, Guo YM. The Combination of Diffusion- and T2-Weighted Imaging in Predicting Deep Myometrial Invasion of Endometrial Cancer: A Systematic Review and Meta-Analysis. J Comput Assist Tomogr 2015;39(5):661-73. doi: 10.1097/RCT.0000000000000280.
Liu D, Yang L, Du D, Zheng T, Liu L, Wang Z, Du J, Dong Y, Yi H, Cui Y. Multi-Parameter MR Radiomics Based Model to Predict 5-Year Progression-Free Survival in Endometrial Cancer. Front Oncol 2022;12:813069. doi: 10.3389/fonc.2022.813069.
In the section “Ovarian cancer”
13. In this section, the authors should add the following references about the discriminating capabilities of DW-MRI
He M, Song Y, Li H, Lu J, Li Y, Duan S, Qiang J. Histogram Analysis Comparison of Monoexponential, Advanced Diffusion-Weighted Imaging, and Dynamic Contrast-Enhanced MRI for Differentiating Borderline From Malignant Epithelial Ovarian Tumors. J Magn Reson Imaging 2020;52(1):257-268.
Türkoğlu S, Kayan M. Differentiation between benign and malignant ovarian masses using multiparametric MRI. Diagn Interv Imaging 2020;101(3):147-155. doi: 10.1016/j.diii.2020.01.006.
Derlatka P, Grabowska-Derlatka L, Halaburda-Rola M, Szeszkowski W, Czajkowski K. The Value of Magnetic Resonance Diffusion-Weighted Imaging and Dynamic Contrast Enhancement in the Diagnosis and Prognosis of Treatment Response in Patients with Epithelial Serous Ovarian Cancer. Cancers 2022;14(10):2464. doi: 10.3390/cancers14102464.
In the section “Pitfalls”
14 In this section, the authors should discuss the lower reproducibility of ADC during chemotherapy of ovarian cancer compared to SUV values and add the following reference
Crombé A, Gauquelin L, Nougaret S, Chicart M, Pulido M, Floquet A, Guyon F, Croce S, Kind M, Cazeau AL. Diffusion-weighted MRI and PET/CT reproducibility in epithelial ovarian cancers during neoadjuvant chemotherapy. Diagn Interv Imaging 2021;102(10):629-639. doi: 10.1016/j.diii.2021.05.007.
References
15. Please add the following references
Di Paola V, Perillo F, Gui B, Russo L, Pierconti F, Fiorentino V, Autorino R, Ferrandina G, Valentini V, Scambia G, Manfredi R. Detection of parametrial invasion in women with uterine cervical cancer using diffusion tensor imaging at 1.5T MRI. Diagn Interv Imaging 2022;doi: 10.1016/j.diii.2022.05.005.
Abdel Wahab C, Jannot AS, Bonaffini PA, Bourillon C, Cornou C, Lefrère-Belda MA, Bats AS, Thomassin-Naggara I, Bellucci A, Reinhold C, Fournier LS. Diagnostic Algorithm to Differentiate Benign Atypical Leiomyomas from Malignant Uterine Sarcomas with Diffusion-weighted MRI. Radiology 2020;297(3):E347.
Lin Y, Wu RC, Huang YL, Chen K, Tseng SC, Wang CJ, Chao A, Lai CH, Lin G. Uterine fibroid-like tumors: spectrum of MR imaging findings and their differential diagnosis. Abdom Radiol 2022;47(6):2197-2208. doi: 10.1007/s00261-022-03431-6.
Lefebvre TL, Ueno Y, Dohan A, Chatterjee A, Vallières M, Winter-Reinhold E, Saif S, Levesque IR, Zeng XZ, Forghani R, Seuntjens J, Soyer P, Savadjiev P, Reinhold C. Development and Validation of Multiparametric MRI-based Radiomics Models for Preoperative Risk Stratification of Endometrial Cancer. Radiology 2022; doi: 10.1148/radiol.212873.
Bereby-Kahane M, Dautry R, Matzner-Lober E, Cornelis F, Sebbag-Sfez D, Place V, Mezzadri M, Soyer P, Dohan A. Prediction of tumor grade and lymphovascular space invasion in endometrial adenocarcinoma with MR imaging-based radiomic analysis. Diagn Interv Imaging 2020;101(6):401-411. doi: 10.1016/j.diii.2020.01.003.
Satta S, Dolciami M, Celli V, Di Stadio F, Perniola G, Palaia I, Pernazza A, Della Rocca C, Rizzo S, Catalano C, Capuani S, Manganaro L. Quantitative diffusion and perfusion MRI in the evaluation of endometrial cancer: validation with histopathological parameters. Br J Radiol 2021;94(1125):20210054. doi: 10.1259/bjr.20210054.
Deng L, Wang QP, Chen X, Duan XY, Wang W, Guo YM. The Combination of Diffusion- and T2-Weighted Imaging in Predicting Deep Myometrial Invasion of Endometrial Cancer: A Systematic Review and Meta-Analysis. J Comput Assist Tomogr 2015;39(5):661-73. doi: 10.1097/RCT.0000000000000280.
Liu D, Yang L, Du D, Zheng T, Liu L, Wang Z, Du J, Dong Y, Yi H, Cui Y. Multi-Parameter MR Radiomics Based Model to Predict 5-Year Progression-Free Survival in Endometrial Cancer. Front Oncol 2022;12:813069. doi: 10.3389/fonc.2022.813069.
He M, Song Y, Li H, Lu J, Li Y, Duan S, Qiang J. Histogram Analysis Comparison of Monoexponential, Advanced Diffusion-Weighted Imaging, and Dynamic Contrast-Enhanced MRI for Differentiating Borderline From Malignant Epithelial Ovarian Tumors. J Magn Reson Imaging 2020;52(1):257-268.
Türkoğlu S, Kayan M. Differentiation between benign and malignant ovarian masses using multiparametric MRI. Diagn Interv Imaging 2020;101(3):147-155. doi: 10.1016/j.diii.2020.01.006.
Derlatka P, Grabowska-Derlatka L, Halaburda-Rola M, Szeszkowski W, Czajkowski K. The Value of Magnetic Resonance Diffusion-Weighted Imaging and Dynamic Contrast Enhancement in the Diagnosis and Prognosis of Treatment Response in Patients with Epithelial Serous Ovarian Cancer. Cancers 2022;14(10):2464. doi: 10.3390/cancers14102464.
Crombé A, Gauquelin L, Nougaret S, Chicart M, Pulido M, Floquet A, Guyon F, Croce S, Kind M, Cazeau AL. Diffusion-weighted MRI and PET/CT reproducibility in epithelial ovarian cancers during neoadjuvant chemotherapy. Diagn Interv Imaging 2021;102(10):629-639. doi: 10.1016/j.diii.2021.05.007.
16. In general, the references are poorly presented. It is possible that the authors believe that Endnote /Zotero can make the job, but this is wrong.
The authors should pay attention to the mdpi style and present their references as it should be. In addition, in some references, journal names are not reported with their correct abbreviated names.
The authors should avoid using references that are not easily accessible (Egyptian Journal of Radiology and Nuclear Medicine) or reusing review articles (Duarte, A.L., J.L. Dias, and T.M. Cunha, Pitfalls of diffusion-weighted imaging of the female pelvis. Radiologia brasileira, 2018. 51(1): 489 p. 37-44.).
Figures.
17. In general, the figures are poorly presented. The authors should make figures with several images labelled A, B, C and so on.
Figures are made of several images which are of different sizes and often not aligned. A more careful presentation is needed. In addition, the number of figures is quite high. Instead of presenting several cases, the authors should focus on the added value of DW-MRI.
In addition, a case with IVIM imaging should be welcome, should it be possible for the authors.
Author Response
In the review, the authors discuss the use of diffusion-weighted magnetic resonance imaging (DW-MRI) in the elective field of gynecological imaging. In general, this a well written review article that encompass the current applications of DW-MRI.
I have the following comments and suggest a few amendments.
The authors should focus their effort on three points: (i), an update of the references; (ii), introduce the concept of IVIM and DTI, which are both part of DW-MRI and have already some applications and interesting results in the field of gynecological imaging; and (iii), a better and more careful presentation of the figures.
In general, the authors should use major and recent references and add a substantial number of references to update their review.
Author’s response: We would like to thank you reviewers for your time to review the paper and valuable suggestions. We have updated the manuscript to the best of our ability.
Author’s response:
As minor comments (but to consider)
- Simple summary: Missing. Please add.
Author’s response:
Short summary is provided under conclusions since its review article.
- Key words: Missing, please add Keywords using MeSH index terms.
Abstract.
Author’s response: Added
- Dermoid is too vague.
Authors response: Role of DWI for dermoid lies in radiologists awareness about the fact that dermoid can show variable diffusion restriction even if benign to avoid misdiagnosis of malignant transformation.
- Spell out ADC.
Main text.
Introduction
Author’s response: Spelled out in the abstract and when first used in the beginning of the manuscript.
- Line 41-42. Relevant references are missing after this sentence “It helps detect and characterize benign lesions, such as tubo-ovarian abscesses and endometriomas, and malignant lesions, such as endometrial, cervical, and ovarian cancers.
Author’s response: The references for this sentence are after the next sentence as they were same to avoid repetition.
- Lines 56-63. The second section of “Principles of DWI” has no references at all. The authors should have relevant references.
In the section “Principles of DWI”
Author’s response: Rechecked and added.
- The authors should briefly describe IVIM and diffusion tensor imaging which are part of DW-MRI and add the following recent references
Dolciami M, Capuani S, Celli V, Maiuro A, Pernazza A, Palaia I, Di Donato V, Santangelo G, Rizzo SMR, Ricci P, Della Rocca C, Catalano C, Manganaro L. Intravoxel Incoherent Motion (IVIM) MR Quantification in Locally Advanced Cervical Cancer (LACC): Preliminary Study on Assessment of Tumor Aggressiveness and Response to Neoadjuvant Chemotherapy. J Pers Med 2022;12(4):638. doi: 10.3390/jpm12040638.
Di Paola V, Perillo F, Gui B, Russo L, Pierconti F, Fiorentino V, Autorino R, Ferrandina G, Valentini V, Scambia G, Manfredi R. Detection of parametrial invasion in women with uterine cervical cancer using diffusion tensor imaging at 1.5T MRI. Diagn Interv Imaging 2022;doi: 10.1016/j.diii.2022.05.005.
Author’s response Updated and added with Di Paola reference.
Dolciami reference and DTI has been already present in the paper as ref [34].
IVIM and DTI are described further down in the article.
- In the section titled Clinical applications, the authors should add the following two references for further discussion
Abdel Wahab C, Jannot AS, Bonaffini PA, Bourillon C, Cornou C, Lefrère-Belda MA, Bats AS, Thomassin-Naggara I, Bellucci A, Reinhold C, Fournier LS. Diagnostic Algorithm to Differentiate Benign Atypical Leiomyomas from Malignant Uterine Sarcomas with Diffusion-weighted MRI. Radiology 2020;297(3):E347.
Lin Y, Wu RC, Huang YL, Chen K, Tseng SC, Wang CJ, Chao A, Lai CH, Lin G. Uterine fibroid-like tumors: spectrum of MR imaging findings and their differential diagnosis. Abdom Radiol 2022;47(6):2197-2208. doi: 10.1007/s00261-022-03431-6.
In the section “Cervical carcinoma”
Author’s response Added and updated.
9 The authors should discuss the role of radiomics from ADC maps and add the following reference
Yamada I, Oshima N, Miyasaka N, Wakana K, Wakabayashi A, Sakamoto J, Saida Y, Tateishi U, Kobayashi D. Texture Analysis of Apparent Diffusion Coefficient Maps in Cervical Carcinoma: Correlation with Histopathologic Findings and Prognosis. Radiol Imaging Cancer 2020 ;2(3):e190085. doi: 10.1148/rycan.2020190085.
Author’s response This is already mentioned in the paper as ref 32.
In the section “Endometrial cancer”
- In this section, the authors should discuss the application of DW-MRI for determining lymphovascular space invasion (LVSI)
The authors should add the following recent reference
Lefebvre TL, Ueno Y, Dohan A, Chatterjee A, Vallières M, Winter-Reinhold E, Saif S, Levesque IR, Zeng XZ, Forghani R, Seuntjens J, Soyer P, Savadjiev P, Reinhold C. Development and Validation of Multiparametric MRI-based Radiomics Models for Preoperative Risk Stratification of Endometrial Cancer. Radiology 2022; doi: 10.1148/radiol.212873.
Author’s response Added and updated
- They should also discuss that one study did not find added value for DW-MRI in the prediction of high grade and LVSI compared to T2-weighted images and add the following reference
Bereby-Kahane M, Dautry R, Matzner-Lober E, Cornelis F, Sebbag-Sfez D, Place V, Mezzadri M, Soyer P, Dohan A. Prediction of tumor grade and lymphovascular space invasion in endometrial adenocarcinoma with MR imaging-based radiomic analysis. Diagn Interv Imaging 2020;101(6):401-411. doi: 10.1016/j.diii.2020.01.003.
Author’s response added and updated
- They should also discuss with more details the role of DW-MRI for determining tumor aggressiveness, myometrial invasion and progression free survival with the following three references
Satta S, Dolciami M, Celli V, Di Stadio F, Perniola G, Palaia I, Pernazza A, Della Rocca C, Rizzo S, Catalano C, Capuani S, Manganaro L. Quantitative diffusion and perfusion MRI in the evaluation of endometrial cancer: validation with histopathological parameters. Br J Radiol 2021;94(1125):20210054. doi: 10.1259/bjr.20210054.
Deng L, Wang QP, Chen X, Duan XY, Wang W, Guo YM. The Combination of Diffusion- and T2-Weighted Imaging in Predicting Deep Myometrial Invasion of Endometrial Cancer: A Systematic Review and Meta-Analysis. J Comput Assist Tomogr 2015;39(5):661-73. doi: 10.1097/RCT.0000000000000280.
Liu D, Yang L, Du D, Zheng T, Liu L, Wang Z, Du J, Dong Y, Yi H, Cui Y. Multi-Parameter MR Radiomics Based Model to Predict 5-Year Progression-Free Survival in Endometrial Cancer. Front Oncol 2022;12:813069. doi: 10.3389/fonc.2022.813069.
Author’s response: Added and updated. The article by Deng et al doesn’t talk about DWI so, it’s not used.
In the section “Ovarian cancer”
- In this section, the authors should add the following references about the discriminating capabilities of DW-MRI
He M, Song Y, Li H, Lu J, Li Y, Duan S, Qiang J. Histogram Analysis Comparison of Monoexponential, Advanced Diffusion-Weighted Imaging, and Dynamic Contrast-Enhanced MRI for Differentiating Borderline From Malignant Epithelial Ovarian Tumors. J Magn Reson Imaging 2020;52(1):257-268.
Türkoğlu S, Kayan M. Differentiation between benign and malignant ovarian masses using multiparametric MRI. Diagn Interv Imaging 2020;101(3):147-155. doi: 10.1016/j.diii.2020.01.006.
Derlatka P, Grabowska-Derlatka L, Halaburda-Rola M, Szeszkowski W, Czajkowski K. The Value of Magnetic Resonance Diffusion-Weighted Imaging and Dynamic Contrast Enhancement in the Diagnosis and Prognosis of Treatment Response in Patients with Epithelial Serous Ovarian Cancer. Cancers 2022;14(10):2464. doi: 10.3390/cancers14102464.
Author’s response Added and updated
In the section “Pitfalls”
14 In this section, the authors should discuss the lower reproducibility of ADC during chemotherapy of ovarian cancer compared to SUV values and add the following reference
Crombé A, Gauquelin L, Nougaret S, Chicart M, Pulido M, Floquet A, Guyon F, Croce S, Kind M, Cazeau AL. Diffusion-weighted MRI and PET/CT reproducibility in epithelial ovarian cancers during neoadjuvant chemotherapy. Diagn Interv Imaging 2021;102(10):629-639. doi: 10.1016/j.diii.2021.05.007.
References
Author’s response: Added and updated accordingly.
- Please add the following references
Di Paola V, Perillo F, Gui B, Russo L, Pierconti F, Fiorentino V, Autorino R, Ferrandina G, Valentini V, Scambia G, Manfredi R. Detection of parametrial invasion in women with uterine cervical cancer using diffusion tensor imaging at 1.5T MRI. Diagn Interv Imaging 2022;doi: 10.1016/j.diii.2022.05.005.
Abdel Wahab C, Jannot AS, Bonaffini PA, Bourillon C, Cornou C, Lefrère-Belda MA, Bats AS, Thomassin-Naggara I, Bellucci A, Reinhold C, Fournier LS. Diagnostic Algorithm to Differentiate Benign Atypical Leiomyomas from Malignant Uterine Sarcomas with Diffusion-weighted MRI. Radiology 2020;297(3):E347.
Lin Y, Wu RC, Huang YL, Chen K, Tseng SC, Wang CJ, Chao A, Lai CH, Lin G. Uterine fibroid-like tumors: spectrum of MR imaging findings and their differential diagnosis. Abdom Radiol 2022;47(6):2197-2208. doi: 10.1007/s00261-022-03431-6.
Lefebvre TL, Ueno Y, Dohan A, Chatterjee A, Vallières M, Winter-Reinhold E, Saif S, Levesque IR, Zeng XZ, Forghani R, Seuntjens J, Soyer P, Savadjiev P, Reinhold C. Development and Validation of Multiparametric MRI-based Radiomics Models for Preoperative Risk Stratification of Endometrial Cancer. Radiology 2022; doi: 10.1148/radiol.212873.
Bereby-Kahane M, Dautry R, Matzner-Lober E, Cornelis F, Sebbag-Sfez D, Place V, Mezzadri M, Soyer P, Dohan A. Prediction of tumor grade and lymphovascular space invasion in endometrial adenocarcinoma with MR imaging-based radiomic analysis. Diagn Interv Imaging 2020;101(6):401-411. doi: 10.1016/j.diii.2020.01.003.
Satta S, Dolciami M, Celli V, Di Stadio F, Perniola G, Palaia I, Pernazza A, Della Rocca C, Rizzo S, Catalano C, Capuani S, Manganaro L. Quantitative diffusion and perfusion MRI in the evaluation of endometrial cancer: validation with histopathological parameters. Br J Radiol 2021;94(1125):20210054. doi: 10.1259/bjr.20210054.
Deng L, Wang QP, Chen X, Duan XY, Wang W, Guo YM. The Combination of Diffusion- and T2-Weighted Imaging in Predicting Deep Myometrial Invasion of Endometrial Cancer: A Systematic Review and Meta-Analysis. J Comput Assist Tomogr 2015;39(5):661-73. doi: 10.1097/RCT.0000000000000280.
Liu D, Yang L, Du D, Zheng T, Liu L, Wang Z, Du J, Dong Y, Yi H, Cui Y. Multi-Parameter MR Radiomics Based Model to Predict 5-Year Progression-Free Survival in Endometrial Cancer. Front Oncol 2022;12:813069. doi: 10.3389/fonc.2022.813069.
He M, Song Y, Li H, Lu J, Li Y, Duan S, Qiang J. Histogram Analysis Comparison of Monoexponential, Advanced Diffusion-Weighted Imaging, and Dynamic Contrast-Enhanced MRI for Differentiating Borderline From Malignant Epithelial Ovarian Tumors. J Magn Reson Imaging 2020;52(1):257-268.
Türkoğlu S, Kayan M. Differentiation between benign and malignant ovarian masses using multiparametric MRI. Diagn Interv Imaging 2020;101(3):147-155. doi: 10.1016/j.diii.2020.01.006.
Derlatka P, Grabowska-Derlatka L, Halaburda-Rola M, Szeszkowski W, Czajkowski K. The Value of Magnetic Resonance Diffusion-Weighted Imaging and Dynamic Contrast Enhancement in the Diagnosis and Prognosis of Treatment Response in Patients with Epithelial Serous Ovarian Cancer. Cancers 2022;14(10):2464. doi: 10.3390/cancers14102464.
Crombé A, Gauquelin L, Nougaret S, Chicart M, Pulido M, Floquet A, Guyon F, Croce S, Kind M, Cazeau AL. Diffusion-weighted MRI and PET/CT reproducibility in epithelial ovarian cancers during neoadjuvant chemotherapy. Diagn Interv Imaging 2021;102(10):629-639. doi: 10.1016/j.diii.2021.05.007.
Author’s response: Added
- In general, the references are poorly presented. It is possible that the authors believe that Endnote /Zotero can make the job, but this is wrong.
The authors should pay attention to the mdpi style and present their references as it should be. In addition, in some references, journal names are not reported with their correct abbreviated names.
The authors should avoid using references that are not easily accessible (Egyptian Journal of Radiology and Nuclear Medicine) or reusing review articles (Duarte, A.L., J.L. Dias, and T.M. Cunha, Pitfalls of diffusion-weighted imaging of the female pelvis. Radiologia brasileira, 2018. 51(1): 489 p. 37-44.).
Author’s response: References are updated.
Figures.
- In general, the figures are poorly presented. The authors should make figures with several images labelled A, B, C and so on.
Figures are made of several images which are of different sizes and often not aligned. A more careful presentation is needed. In addition, the number of figures is quite high. Instead of presenting several cases, the authors should focus on the added value of DW-MRI.
In addition, a case with IVIM imaging should be welcome, should it be possible for the authors.
Author’s response: It is difficult to depict information based on DWI images alone so relevant conventional and DWI images are provided. Images are updated again and journal support team mentioned that they will be rearranging the images according to journal format when accepted. We could not get IVIM image.
Reviewer 2 Report
Thank you very much for the opportunity to review the paper titled “Utility of the Diffusion Weighted Sequence in Gynecological Imaging”. Authors raised an important role of imaging modality in gynecology. However, some major changes are needed before the accepting this paper to publication.
1) Please rewrite the whole manuscript according to the PRISMA guideline, which is recommended for review articles.
2) The title should indicate that this is a review, thus please rewrite the title of the manuscript.
3) Please check the whole manuscript for double space or no space.
4) The reference number “3” should be before 4 in the Introduction section
5) Please explain the abbreviations used for the first time in the main text, not further. For example (but not limited to: In the introduction the explanation of abbreviation of MRI, there is no explanation of “CT”; “ROI” and “DTI” are explained in the text later, not when they first appear).
6) Once authors write “T2-weighted” once “T2 weighted”. The same with “b-value” and “cut-off”. Please indicate the homogenous nomenclature in the whole manuscript.
7) Figures should be with the same size and in order. In this form, manuscript looks chaotic.
8) “Authors contributions” section is missing.
9) Please change the reference section according to the journal guideline
10) In reference section: why reference 24 is “invalid citation”?? Please provide changes.
Author Response
Thank you very much for the opportunity to review the paper titled “Utility of the Diffusion Weighted Sequence in Gynecological Imaging”. Authors raised an important role of imaging modality in gynecology. However, some major changes are needed before the accepting this paper to publication.
Author’s response: We would like to thank you reviewers for your time to review the paper and valuable suggestions. We have updated the manuscript to the best of our ability.
- Please rewrite the whole manuscript according to the PRISMA guideline, which is recommended for review articles.
Author’s response: Updated as much as possible
- The title should indicate that this is a review, thus please rewrite the title of the manuscript.
Author’s response: Updated to “Utility of the Diffusion Weighted Sequence in Gynecological Imaging: Review Article”
- Please check the whole manuscript for double space or no space.
Author’s response: Rechecked and updated
- The reference number “3” should be before 4 in the Introduction section
Author’s response: Updated
- Please explain the abbreviations used for the first time in the main text, not further. For example (but not limited to: In the introduction the explanation of abbreviation of MRI, there is no explanation of “CT”; “ROI” and “DTI” are explained in the text later, not when they first appear).
Author’s response: Checked and updated
- Once authors write “T2-weighted” once “T2 weighted”. The same with “b-value” and “cut-off”. Please indicate the homogenous nomenclature in the whole manuscript.
Author’s response: Rechecked and updated
- Figures should be with the same size and in order. In this form, manuscript looks chaotic.
Author’s response: Figures are updated.
- “Authors contributions” section is missing.
Author’s response: Added.
- Please change the reference section according to the journal guideline
Author’s response: Updated according to journal guidelines
- In reference section: why reference 24 is “invalid citation”?? Please provide changes.
Author’s response: Ref 24 is deleted and instead ref 1 and 5 are used.
Round 2
Reviewer 2 Report
No other comments.